

# Motivational strategies and approaches for single and multi-player exergames: a social perspective

Gerry Chan[1], Ali Arya[1], Rita Orji[2] and Zhao Zhao[3]

[1] Carleton School of Information Technology, Carleton University, Ottawa, ON, Canada
[2] Faculty of Computer Science, Dalhousie University, Halifax, NS, Canada
[3] Department of Systems and Computer Engineering, Carleton University, Ottawa, ON, Canada

## ABSTRACT

**Background:** Exergames have attracted the interest of academics, practitioners, and designers, in domains as diverse as health, human-computer interaction, psychology, and information technology. This is primarily because exergames can make the exercise experience more enjoyable and entertaining, and in turn, can increase exercise levels. Despite the many benefits of exergames, they suffer from retention problems. Thus, the objective of this article was to review theories and game elements that have been empirically examined or employed in an attempt to make exergames more motivating so people engage in sustained physical activity (duration of physical activity) in a repeating pattern over time (frequency of physical activity).
**Methodology:** A literature search and narrative review were conducted.
**Results:** Five major theories and elements were prevalent in the exergaming literature: (1) self-determination theory, (2) gamification, (3) competition and cooperation, (4) situational interest, and (5) social interaction. These theories and elements are important for encouraging long-term play and show promise for designing exergames to promote sustained engagement and motivate physical activity. We discuss their strengths and weaknesses throughout the paper.
**Conclusions:** The long-term effectiveness of exergame interventions is unclear mainly because of the limited amount of long-term studies. Better metrics are also needed to evaluate this effectiveness. We also identified particular attention to social factors and group dynamics, such as multi-player exergames and more effective player matchmaking strategies for increasing social connectedness, as a key area of future research.

## INTRODUCTION

A sedentary lifestyle is linked to many health concerns. Research suggests that sedentary behavior—activities that involve extensive amounts of sitting such as television viewing and desk-bound work, are associated with an increased risk of developing metabolic dysfunction, cardiovascular disease, obesity, and lower levels of psychosocial health and well-being (*Tremblay et al., 2010*). In contrast, living an active lifestyle through engaging in regular exercise and physical activity is associated with many health benefits (*Alpert, 2009*).

Corresponding author
Gerry Chan,
gerrychan@cmail.carleton.ca

[1] Exergames are also commonly referred to as active video games, active gaming, movement-controlled video games, or exertion games, in which interaction of the interface requires physical effort (*Yim & Graham, 2007*; *Mueller, Gibbs & Vetere, 2010*).

However, common complaints associated with participation in exercise include factors such as lack of social support, perceived feelings of exhaustion, and inconvenience of environmental conditions (*Myers & Roth, 1997*). As a result, research has increased focused on how to harness the power of technology to support people to be physically active. These efforts have led to the emergence of videogames that require players use a range of active body motions and hence encouraging physical activity in a fun and engaging way. Such games are commonly referred to as "exergames." Exergames[1], which are a combination of exercise and videogames (*Yim & Graham, 2007*), can make doing exercise more enjoyable, and offer a safe, entertaining, and engaging environment to motivate people to participate in physical activity (*Altamimi & Skinner, 2012*). In this paper, we use the term *exergame* as a gamified experience that combines physical activity (traditional exercise which involves "planned, structured, repetitive" movements (*Caspersen, Powell & Christenson, 1985*) or other types of practice that makes the player physically active) using game elements.

Despite the numerous benefits of exergames, they suffer from retention problems (*Graves et al., 2016*; *Rhodes et al., 2019*; *Sun, 2012*). While some games entice players to crave more play-time, other games dissuade players before they reach the next level or the end-goal of a game. The problem of maintaining players' motivation and keeping them actively engaged is commonly referred to as player retention or game sustainability, and has been an important and long-running investigation in the gaming community (*Debeauvais et al., 2010*; *Weber, Mateas & Jhala, 2011*). In this paper, the term "retention" is used interchangeably to describe a player's continued participation in an activity (such as exercise activity, videogame play, or exergame play) over time. While players' participation is the primary goal in this regard, keeping them motivated can be the strongest tool the designers have, although participation is a function of many other parameters. Also, it is possible that players may discontinue playing a game, but continue being active through playing other exergames or participating in other physical activities/sports. Since the designer of any particular game may not know about what happens after the player leaves, the word participation in the context of this paper refers to continued play in the same game. The duration of sustainability would continue until the player reaches the end of the game; but ideally, to keep living an active lifestyle, the player would move onto a new game, repeat play of the same game, until they are able to maintain their exercise routine without needing any motivation eventually, hence forming habitual health behaviours (*Aarts, Paulussen & Schaalma, 1997*).

The purpose of this review is to study the strategies, theories, principles, and practices that are considered by the existing research when designing exergames to increase players' motivation as a means of increasing participation and as such improving retention/ sustainability. This includes the approaches that researchers have employed in an attempt to create a more engaging and enjoyable play experience. Although much research on exergames exists and is ongoing, there remain open questions about how we can keep players motivated over the long-term. The following two questions about exercise and videogames are explored in this paper:

- What social features can be integrated in exergames to motivate and sustain engagement?
- How can existing theories and motivational strategies (particularly from social psychology and personality) be used to optimize the exergame experience and encourage continued play in a sustained and repeated pattern over time?

The research presented in this paper was motivated by these open questions and our observation of limited attention to some motivating factors such as social interaction in existing research.

The field of exergames has been reviewed recently by other researchers, for example: the social effects of exergames on older adults (*Theng et al., 2018*); the effectiveness of gamification in exergames (*Matallaoui et al., 2017*); and the psychological effects of playing exergames (*Lee et al., 2017*). However, there is a lack of investigation and evaluation on social features used in exergames to increase motivation, particularly considering the long-term effects. This review offers a new perspective on the motivational and social aspects of exergame retention. It contributes a multidisciplinary understanding of how to design exergames to motivate players and sustain engagement, through effective use of social features in the design of exergames.

Our specific contributions can be grouped as follows:

- Classification of existing approaches, social features, game elements, and theoretical models to provide a more comprehensive insight into what motivates play. We identified the following five categories:

  ○ Gamification
  ○ Self-Determination Theory (SDT)
  ○ Competition vs. Cooperation
  ○ Social Interaction
  ○ Situational Interest

- Evaluation of the effectiveness of the most common approaches
- Identification of some of the most promising areas of research that can improve the existing approaches:

  ○ Focus on social aspects
  ○ Longitudinal studies
  ○ Domain specific evaluation metrics

The remainder of this paper is structured as follows. First, we report on the process employed to conduct the review and the results. Then, we provide an overview of popular motivational approaches and strategies that have been studied in the videogame and exergame literature as they relate to game play followed by an evaluation of the five major categories. Based on our evaluation each category, we identify gaps in the current body of

literature and offer a discussion of directions for future research together with potential research questions.

## LITERATURE REVIEW

### Research approach

To achieve the intended objective of this paper, we reviewed both theoretical and empirical evidence concerning the motivational aspects of exergame play. We employed a snowball method (*Wohlin, 2014*) where we started with some initial set and used their references, both backward and forward iterations, until we no longer found significant new work.

For our literature search, we used Google Scholar as our first data source. The terms *active video games* and *exergames*, together with the terms *enjoyment* ($n = 4,150$), *engagement* ($n = 5,670$), and *motivation* ($n = 7,540$) were all included in the search process that used the full body of articles. We also searched PubMed, PsycINFO, Scopus, Springer, and the ACM Digital Library using the same search terms. This ensures good coverage of empirical studies on exergames across various fields including Human-Computer Interaction (HCI), technology, psychology, computer games, and other related disciplines. These disciplines were selected for the search because they are the home to most research on games, gamification, and understanding the player experience. Research projects were included when researchers (a) examined the effects of videogame play—predominantly exergames, (b) explored and measured some outcome variable related to motivation (e.g., enjoyment and engagement) and (c) studied the psychology of the player and playing behaviors, particularly related to continued play. Papers with key terms such as "active video games," "exergames," "motion-based games," and "motivation" were all included in the analysis. Included in the search were also reviews of research specifically conducted on exergames (*Altamimi & Skinner, 2012*; *Kari, 2015*; *Kooiman & Sheehan, 2015a*; *Lee et al., 2017*; *Lyons, 2014*; *Matallaoui et al., 2017*; *Mellecker, Lyons & Baranowski, 2013*; *Peng, Crouse & Lin, 2013*). Reference sections of review papers were chosen for inclusion in this review when there was some kind of theoretical component(s) or outcome measure(s) related to the psychology of motivation and social interactions.

To ensure that we cover a wide range of concepts and theories related to videogame play, in addition to the initial keyword search, we also examined other literature pertaining to all types of videogames (e.g., mobile, single-player, multiplayer online role-playing, and many others). Then, because of our particular interest in the social aspects of play, we focused mainly on multiplayer videogames and social motivations for playing exergames. Not all general topics in videogames are significant or applicable in the special case of exergames. Thus, we further explored literature that we believed were applicable to exergames and relevant for understanding the effects of continued play which included motivational theories, as well as other related theories that have been researched in videogames. During the database search process we included articles that discussed the design and evaluation of motivational and social approaches. We excluded important topics such as aesthetics that are relevant to all games (and other applications) but are not specifically significant for exergames or are not relevant for the current context of this review. Papers describing the design and development of exergames without an evaluation

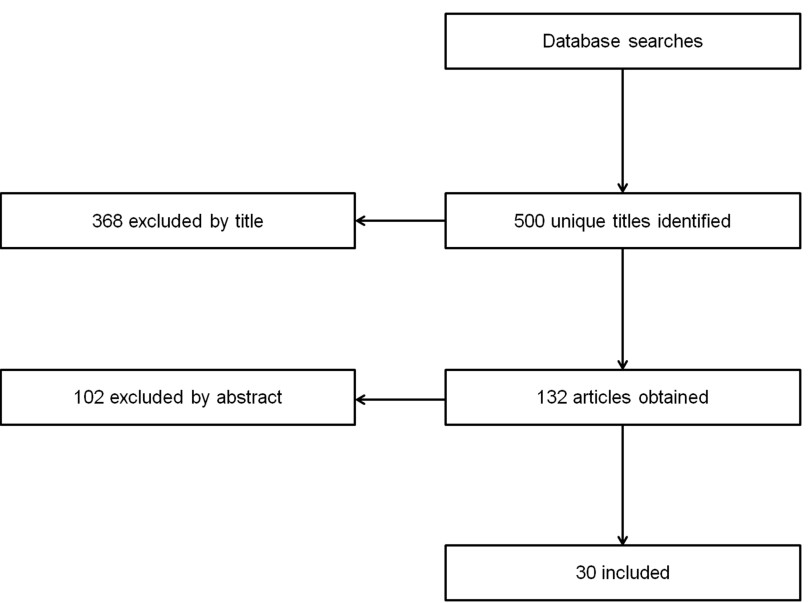

**Figure 1 Included study identification process.**

component and without psychological theory were also excluded. Data extractions in accordance with the research questions posed in this review were conducted concurrently with deciding on the papers to include or exclude. We also searched in specific journals (e.g., Games for Health Journal) and conferences (e.g., ACM CHIPlay) that are likely to include more papers on the topic.

After scanning through the titles, abstracts and keywords of the articles we gathered from our initial search, 90 papers were selected as starting set (seed) of snowball based on relevance, date, and citation count. We made sure the seed papers were as diverse as possible with regards to authors, publishers, and topic. We performed backward and forward snowballing, reviewing papers in the reference lists of our articles and those that have cited our articles. The results of snowball yielded more than 500 unique titles of which 132 papers and books were deemed relevant (meeting our inclusion criteria) following an examination of the titles, abstracts, and full body text. Out of these 132 references, some of them were theoretical papers ($n = 18$) while some described actual user studies ($n = 83$), and many were review papers ($n = 31$). Figure 1 summarizes the study identification process.

Among the exergame studies that were deemed to fit the scope of this review, we identified 30 articles (17 journal articles, 12 conference papers, one book chapter) as typical examples of work on the subject and included in a more detailed discussion (listed in Table 5) while all papers were reviewed. The collection of 30 typical papers was published from 2009 to May 2019 and demonstrates the most common trends in exergame motivational strategies. For example, many researchers have studied the motivational and social aspects of competitive and cooperative exergame play employing similar methods, outcome measures and take on the same theoretical foundations (*Hsieh & Peng, 2012*; *Song et al., 2013*; *Staiano, Abraham & Calvert, 2012*).

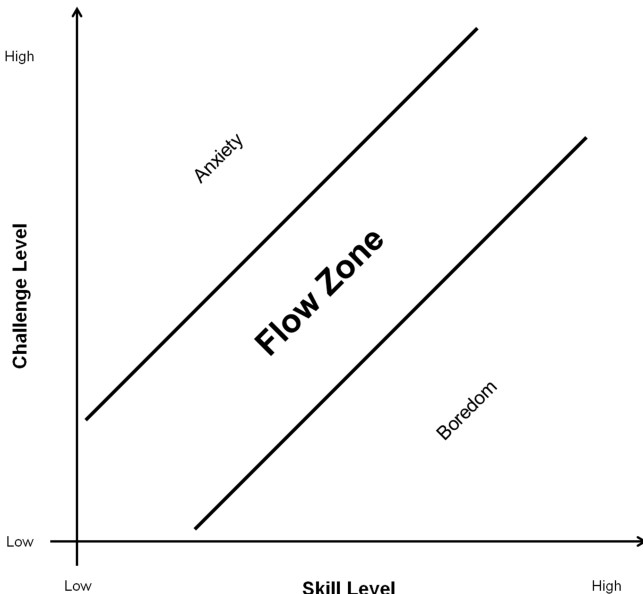

**Figure 2 Model of the Flow zone in relation to challenge and skill level adapted from *Nakamura & Csikszentmihalyi (2014)*.**

While in the survey we particularly focused on the 30 selected articles, we included other examples in our references and mentioned them in the appropriate section. We also included theoretical and survey papers that reviewed related topics such as increasing the level of enjoyment experienced in exergames (*Lyons, 2014*; *Mellecker, Lyons & Baranowski, 2013*). Enjoyment was included because it has been associated with continued motivation and social aspects in exergames.

## Motivation and gameplay

Research shows that *motivation* is the most important force for determining a player's desire for continued play (*Stankevicius et al., 2015*) and in an exergame context, consideration of self-efficacy is particularly important for maintaining motivation over time through setting achievable goals (*Macvean & Robertson, 2013*). Motivation can be defined as the driving force behind all the actions of an individual (*Rabideau, 2005*) and most researchers agree that most motivation theories attempt to explain three interrelated aspects of human behavior: (1) the *choice* of a particular action (2) the *persistence* of a particular action, and (3) the *effort* expended on a particular action (*Dörnyei, 2000*). One theory that is commonly associated with motivation is the "flow" theory (*Csikszentmihalyi, 1977*) which is characterized as a mental state marked by complete absorption and high concentration of an activity. *Chen (2007)* argues that a well-designed videogame will motivate repetitive play and "transports" its players to their personal "Flow Zones"—where a person's abilities are matched by a challenge, delivering genuine feelings of pleasure and happiness. Moreover, games should adapt to the player's skills in order to keep them in the *flow zone*. If the challenge exceeds the person's abilities, "anxiety" will be experienced, whereas too little challenge leads to "boredom" (Fig. 2). In the context of exergames, *Huang et al. (2018)* found that competitive individuals have an enhanced need for feelings

**Table 1 Factors for increasing enjoyment in health games adapted from *Crutzen, van't Riet & Short (2016)*.**

| Factor | Definition | Design recommendations |
|---|---|---|
| Competence* | The perception of increasing skills (*Ryan, Scott Rigby & Przybylski, 2006*). | Provide feedback, challenge and rewards (*Lyons, 2014*). |
| Narrative Transportation | A process in which someone is mentally "transported" away from the physical world into the imaginary world presented in the form of a story (*Green & Brock, 2000*). | Characters provide the driving force or a narrative (*Lu et al., 2012*) and the ethnic similarity between game characters and players enhances immersion (*Lu et al., 2012*). |
| Relevance | "Closely connected or appropriate to the matter at hand" (as cited in *Crutzen, van't Riet & Short (2016)*). Both "game world" and "real world" relevance is important for facilitating enjoyment. | Self-identification with game characters. Games are most intrinsically motivating when players' experience of themselves is congruent with their conceptions of their ideal selves during play (*Przybylski, Weinstein & Murayama, 2012*). |

Note:
* *Crutzen, van't Riet & Short (2016)* hypothesized that competence is more important for influencing enjoyment, even more so than autonomy and relatedness.

of achievement and are focused on overcoming challenges, and thus, can easily experience flow. The researchers also found that perceived challenge and exercise enjoyment were both positively related to the experience of flow.

Furthermore, *Crutzen, van't Riet & Short (2016)* aimed to clarify the concept of *enjoyment* among other related constructs such as fun and engagement in the context of games for promoting health behaviors. The researchers reviewed empirical studies and found that concepts such as flow, engagement, and fun are associated with enjoyment. Engagement which refers to (the level of motivation that a player displays in gameplay reflecting a psychological process (*Caroux et al., 2015*)) is related to immersion which refers to (a state of high motivation to play the game, while retaining some awareness of one's surroundings (*Baños et al., 2004*)) and presence which refers to (the experience of being personally and physically inside a virtual environment (*Wirth et al., 2007*)). Both engagement and enjoyment are important for keeping high levels of motivation and thus, are key factors for designing games that aim to promote health behaviors. *Crutzen, van't Riet & Short (2016)* concluded their review suggesting that other researchers use enjoyment, and enjoyment only, to refer to the action or state of deriving gratification from a game, and identified three important factors for increasing the level of enjoyment while designing health games: (1) competence, (2) narrative transportation, and (3) relevance (Table 1).

Many researchers have explored the value of videogames for encouraging participation in physical activity (*Altamimi & Skinner, 2012*; *Whitehead et al., 2010*) and have identified principles for designing successful exergames (*Mandryk, Gerling & Stanley, 2014*; *Yim & Graham, 2007*). There is also a substantial amount of research regarding the motivational aspects of videogame play (*Kooiman & Sheehan, 2014*; *Przybylski, Rigby & Ryan, 2010*; *Ryan, Scott Rigby & Przybylski, 2006*; *Yee, 2007*). *Macvean & Robertson (2013)* suggest that in order to sustain exergame play longitudinally, designers need to consider how to positively influence participants' motivation. *Lyons et al. (2014)* and *Lyons (2014)* further suggest that maintaining high levels of enjoyment, which is a component of motivation, is essential to promoting continued game-play and exercise adherence.

However, not all games are well-designed and thus, are unable to keep players motivated. *Dias & Martinho (2011)* suggest that all videogames eventually "lose their

charm and the interest of the player" and by then, players should have already experienced all the key features, yet many game stop being fun before they have the chance to do that. This occurs because of two reasons: (1) players do not relate to the challenge they are assigned, and (2) players do not appreciate the rewards the game is giving them in return for their efforts (*Chen, 2007*). Research shows that factors such as lack of time to play, loss of interest because players no longer find the current game fun, the novelty of a different game (*Tyack, Wyeth & Johnson, 2016*), unsatisfactory player-to-player matchmaking recommendations for facilitating social connectedness (*Horton, Johnson & Mitchell, 2016*) and player toxicity (unsportsmanlike behaviors that are displayed by a player that are undesired by other players) (*Riegelsberger et al., 2007*), can all result in players quitting a game. As for exergames, there is research suggesting that some exergames may not be able to maintain long-term interest in exercise (*Sun, 2013*) and reasons for not engaging with exergames over a long-term include factors such as games not being useful enough, lack of time, and preference for other forms of exercise (*Kari et al., 2012*). By understanding the factors that attract players and the factors that do not, designers can tailor the design to keep motivation levels high, attract new players, while sustaining already existing ones.

The subsequent sections offer an analysis of some of the most common theories and elements that have been applied to study the motivational effects of exergame play. These are SDT, gamification, competition and cooperation, situational interest, and social interactions. The boundaries between these theories and elements are relatively fluid in that some of them are often studied collectively and share similar properties with each other. For example, the psychological need of relatedness from *Ryan & Deci (2000)* SDT is often studied by means of social interaction, as well as cooperation. Gamification is also often conceptualized with respect to intrinsic and extrinsic motivation which was emerged from the SDT.

## Self-determination theory

Self-determination theory is a macro-theory derived from empirical work on human motivation and personality in social contexts (*Ryan & Deci, 2000*). There are three conditions supporting the SDT: (1) autonomy, (2) competence, and (3) relatedness. Autonomy refers to the perception of being in control of one's own activities and decisions, competence refers to the perception of increasing mastery of skills, and relatedness refers to the perceived development and maintenance of close personal relationships. All three conditions are necessary to foster the most "volitional and high quality" forms of motivation for engaging in activities involving enhanced performance, persistence, and creativity (*Ryan & Deci, 2000*). SDT further assumes that people are "active organisms" and can be viewed in relation to intrinsic (doing an activity simply for the enjoyment of the activity itself, rather than for external rewards or pressures (*Ryan & Deci, 2000*)), and extrinsic (doing an activity in order to attain some separable outcome or value (*Ryan & Deci, 2000*)) motivation represented along a continuum (Fig. 3). In quantitative studies measuring intrinsic motivation, the concept of game enjoyment is commonly associated with an individual's experience of fun and their level of interest (*Mekler et al., 2014*).

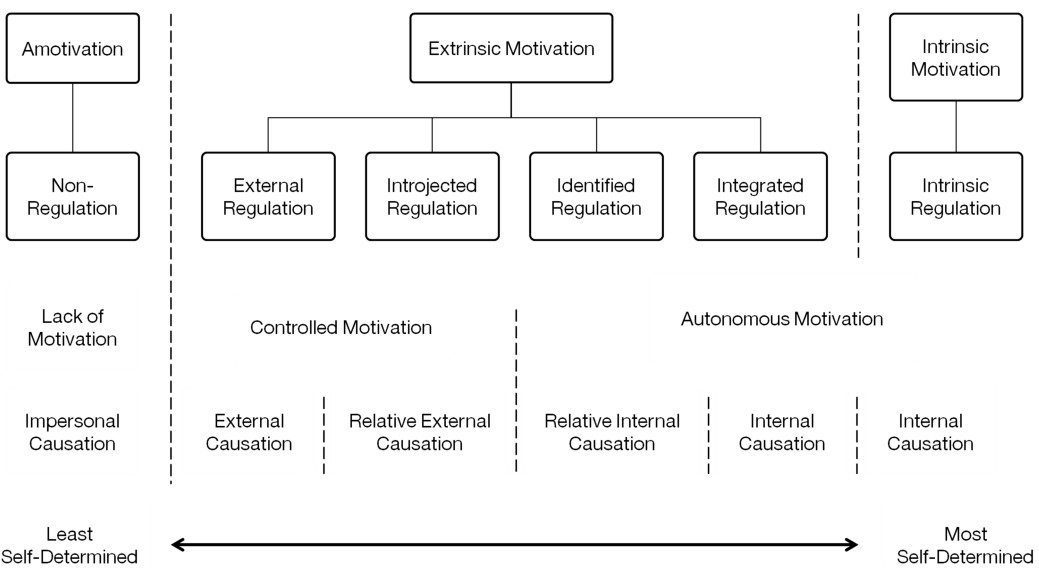

**Figure 3 The self-determination continuum adapted from *Deci & Ryan (2009)* SDT and the facilitation of intrinsic motivation, social development, and well-being.**

From a psychological perspective, *Ryan & Deci (2000)* claim that autonomy, competence, and relatedness can also be conceptualized as essential human needs. In the context of videogames, *Przybylski, Rigby & Ryan (2010)* proposed a motivational model of videogame engagement based on SDT to provide experiences that satisfy universal human needs. According to the model, video games are motivational because they satisfy the need for autonomy, competence, and relatedness. SDT is one of the most prominent theories used to inform the design and evaluation in the exergame literature (*Kooiman & Sheehan, 2015a*; *Lindgaard, 2018*). Researchers such as *Boulos & Yang (2013)* have proposed a framework (Fig. 4) based on SDT for conducting future research particularly on children's motives for participating in games. The researchers recommend that research should aim to understand the types of activities that children already find compelling and motivating, which may include enjoyment or physical appearance or weight loss. The researchers further posit that if players feel competent during play, they are more likely to continue playing because continued feeling of competence will likely lead to intrinsically-motivated behaviors resulting in outcomes such as higher levels of physical activity, enjoyment, and adherence.

Other researchers such as *Lin et al. (2012)* have conducted some empirical work to examine the usefulness of the SDT in the context of exergames and found evidence that exergame features appear to influence basic needs for satisfaction of players. Results showed that several game design strategies and choices available in the game, such as the ability to customize and increase the strength of the avatar, supported the need for autonomy. The ability to dynamically adjust the level of difficulty based on a player's performance and achievement indicators satisfied the need for competence. Additionally, the researchers reported that autonomy-supportive and competence-supportive game features

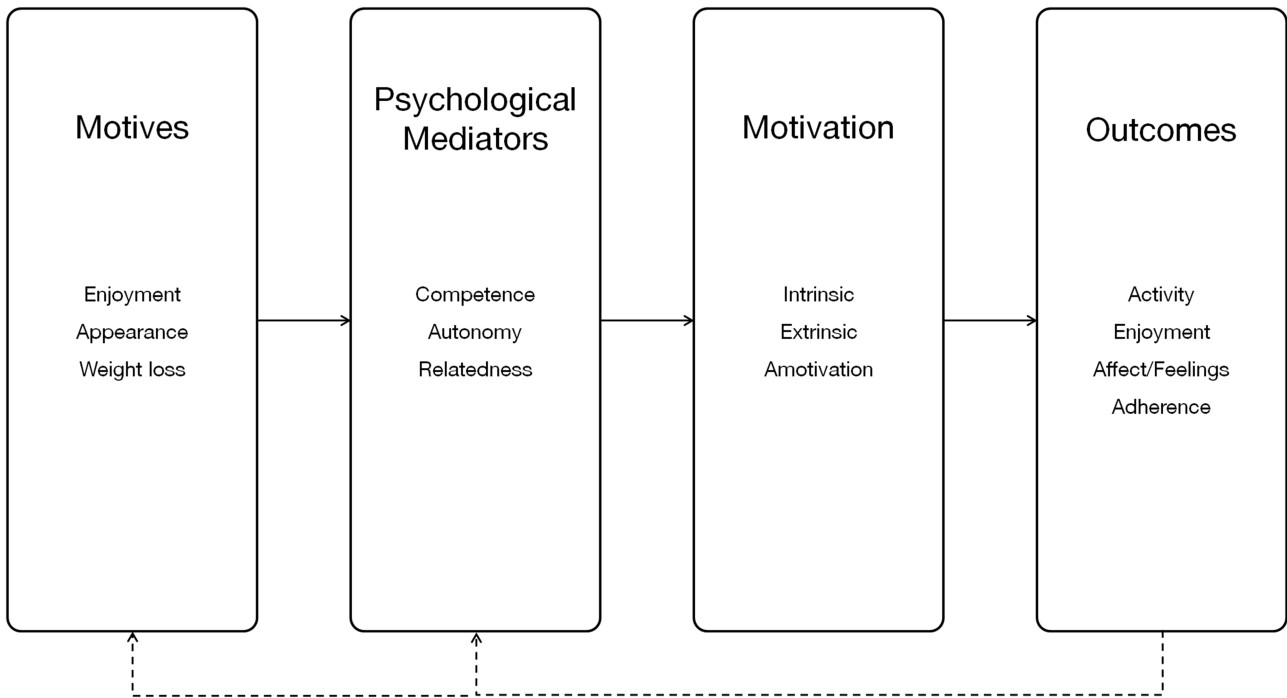

**Figure 4 Relationship between SDT mediators, motivation, outcomes, and motives (*Boulos & Yang, 2013*).**

led to higher levels of game enjoyment, higher levels of motivation for future play, and greater self-efficacy for exercise using the game, and greater game rating. However, no effects of the need satisfaction-supportive game features on players' self-reported effort exerted in gameplay were found. The researchers concluded that "theory-prescribed" game features contribute to the experience of enjoyment, motivation, and engagement through feelings of need satisfaction and recommended that future research intending to harness the power of videogames needs to include game features that support need satisfaction to increase the level of enjoyment, motivation and engagement outcomes.

In a related research, *Osorio, Moffat & Sykes (2012)* have also studied the SDT in exergames, particularly as it relates to a social context. Surveys were conducted to investigate the motivations that encourage participation in exergames and found that the need for both autonomy and relatedness were highly satisfied within a social context, yet the need for competence was lower compared to the other two needs. The researchers speculated that this was because exergaming was perceived as a social activity and that participation in exergames might be motivated by the enjoyment of social interactions rather than competition. More recently, *Limperos & Schmierbach (2016)* examined the relationship between exergame play experiences, enjoyment, and intentions for continued play, and found that player performance directly and indirectly predicts feelings of psychological responses (autonomy, competence, and presence), enjoyment of the experience, and the likelihood for future play. In particular, they reported that players who

achieved better performance experienced greater levels autonomy, presence, and enjoyment. They concluded that performance in exergames is related to the player's psychological experiences that motivate enjoyment and the intention for continued play.

*Staiano et al. (2016)* conducted a 12-week study to examine the potential "transfer effects" (become more physically active outside game play (*Staiano et al., 2016*)) of exergaming on external behaviors such as physical activity levels, screen time and physiological constructs. Overweight or obese adolescents were selected to participate in an exergaming intervention, 60 min sessions of group-based dance sessions three times per week, or a no-treatment care control group. Participants were given the freedom to choose the intensity and game play among a variety of dance-based games. At the end of 12 weeks, participants answered questions regarding media use, self-efficacy toward physical activity and level of intrinsic motivation. The group which participated in the intervention reported high levels of intrinsic motivation to play exergames and high levels of self-efficacy toward physical activity. Results also showed that adolescents highly enjoyed the exergaming experience and intend to continue play in the future. This is one of the first studies in this literature review so far that attempts to transform passive screen time to active screen time. Although evidence for transfer effects on physical activity was not significant, it still provided some support for the idea that videogames can be used to as an effective tool for motivating physical activity.

To summarize, there is an abundance of literature supporting the SDT, particularly at the individual level and researchers continue to examine SDT with respect to the likelihood of exergames informed by SDT to facilitate player retention and prolonged play.

## Gamification

Gamification refers to the application of game principles in non-game contexts (*Deterding et al., 2011*) and is inherent in all exergames. An exergame without gamification would be only exercise. Gamification by itself is not necessarily a social strategy, some game features can consist of social mechanics, such as leaderboard where players are able to see and compare their own performance with other players. How to effectively use gamification features (particularly social features) as a motivator in exergames to sustain long-term play has been the research objective of many studies (*Brauner et al., 2013*; *Hamari & Koivisto, 2015*). Gamification has been applied to a variety of health-related applications to promote a positive influence on health and wellness behaviors in recent years (*Pereira et al., 2014*) and can work by providing a source of motivation for one to engage in physical activity. *Deterding (2012)* argue that in order for gamification to be successful, proper "game design" elements (*Schell, 2008*) must be included, not simply game components. Despite the growing interest in persuasive games and gamified systems for motivating health-related behaviors (*Orji & Moffatt, 2018*; *Orji, Nacke & Di Marco, 2017*), the majority of persuasive games employ a non-tailored approach to their design (*Busch et al., 2015*). Although some researchers are attempting to design better measure instruments to more accurately evaluate user preferences and motivations to use a gamified system (*Tondello et al., 2016*), very little research has been done to design better metrics to evaluate player

preferences and motivations for personalizing the experience specifically in the context of exergames.

*Matallaoui et al. (2017)* reviewed existing empirical research on the deployed motivational affordances and the effectiveness of including gamification features in exergames. The researchers conducted the review based on a framework proposed by *Hamari, Koivisto & Sarsa (2014)* consisting of three major elements: (1) the motivational affordances (i.e., game elements employed in a game system such as progress badges or leaderboards), (2) the psychological outcomes that are induced by the motivational affordances (i.e., the psychological processes such as competition, achievement, and self-expression), and (3) the resulting and pursued behavioral and quasi-medical outcomes (attitudes) such as increased physical activity and measured health benefits. Results revealed positive psychological outcomes such as enjoying the exercise experience, as well as behavioral ones such as decreasing sedentary behaviors and motivating engagement in exercise. However, the researchers also identified some shortcomings in the literature such as the use of a small sample size and non-validated scales, the majority of studies were based on user evaluation rather than control groups and relatively short timeframes, as well as the lack of linkage between theory and practice. The researchers concluded that among the literature reviewed, there were only a few studies that considered proper game design for promoting physical activity. Thus, the researchers recommended that more theory-based studies are needed to examine the effectiveness of different gamification elements and mechanics in exergames.

*Macvean & Robertson (2013)* conducted a 7-week long case study to examine how users' physical activity, motivation and behavior changes over-time. The concept of self-efficacy (how well one can execute courses of action required to deal with prospective situations (*Bandura, 1982*)) was applied within the context of a school based exergame intervention, including how players set goals and manage different levels of difficulty. Results showed that participants were motivated by different factors. While some participants were highly motivated by earning points other players strived to improve performance and setting goals. In particular, differences were found within individual preferences with respect to how points were treated—some were interested in earning points by competing with peers, some enjoyed keeping points hidden from others while some preferred to broadcast their achievements.

*Zhao et al. (2017)* conducted a 70-day study on the effects of wearable-based exergames. In particular, the researchers were interested in whether or not the gradual release of game features can improve engagement over time. Participants were divided into three different groups: Group 1 received a game with basic features, Group 2 received full features (i.e., customization and multiplayer modes), and Group 3 gradually received new features that were unlocked every 10 days. The behavioral patterns of each group were tracked and at the end of the study, participants completed a post-study questionnaire to evaluate their experience during the study. Results showed that the level of engagement in exercise and game-play are highly related, and that the gradual addition of new features increases the amount of application usage and overall physical activity. Results also showed that a

**Table 2 Affordances that facilitate participation in physical activity with respect to four different age categories (*Kappen, Mirza-Babaei & Nacke, 2017*).**

| Age category (years old) | Affordances | |
| --- | --- | --- |
| | Motivate to participate | Continuation over the long-term |
| 18–29 | Badges, progression, goals | Badges, goals, progression |
| 30–49 | Calories, step-counters, progression | Progression, step counters, time |
| 50–64 | Calories, distance travelled, weight loss | Goals, step counters, feedback |
| 65+ | Step-counters, distance travelled, progression | Feedback progression, goals |

gamified exercise experience was more preferable compared to a regular exercise experience.

Furthermore, researchers such as *Geelan et al. (2016)* attempted to increase both the duration and the intensity of exercise activity by designing a system that combines traditional exercise equipment such as exercise bicycles with videogames to provide additional interactive feedback and motivation through exergame components. Results indicate that the addition of augmented reality and videogames to existing exercise routines along with gamified elements, can increase the level of energy expenditure and the amount of time spent exercising compared to non-gamified exercise equipment. However, whether or not this approach is able to hold the attention of players over the long-term is open to question, and thus the researchers identify this as an opportunity for future research to examine the sustainability of exercise activities. In a different study, *Kappen, Mirza-Babaei & Nacke (2017)* explored the effects of gamified motivational affordances (the properties of an object that determine whether and how it can support one's motivational needs (*Zang, 2008*)) for facilitating physical activity preferences and found that various age groups are motivated by different kinds of motivational affordances (Table 2) which consists of both intrinsic and extrinsic elements, as well as feedback elements (Table 3).

In general, the effectiveness and ways to gamify exercise routines continues to be a subject of interest for many game designers and researchers. Research results suggest that a number of elements (both intrinsic and extrinsic rewards) are important for inclusion in the design of exercise interventions and that more studies are needed to examine different gamification elements that can be used to encourage long-term play.

## Competition and cooperation

Competition and cooperation (same team playing together to achieve a shared/common objective) are features offered in traditional videogames and are important elements in game enjoyment and influences player's choice of games (*Vorderer, Hartmann & Klimmt, 2003*). Previous research has shown that competition and cooperation both have an influence on exercise performance, motivation, and enjoyment (*Peng & Crouse, 2013*; *Staiano, Abraham & Calvert, 2013*). *Marker & Staiano (2015)* conducted a review of existing literature on the effects of social exergaming, specifically how competitive and cooperative components of social play influence physical activity and motivation. They

**Table 3 Differentiating gamified motivational affordances and feedback elements (*Kappen, Mirza-Babaei & Nacke, 2017*).**

| | Motivational elements | | |
|---|---|---|---|
| Gamified motivational affordances | Intrinsic | Goals<br>Challenges<br>Progression<br>Achievements | Choices/options<br>Quests<br>Social sharing |
| | Extrinsic | Badges<br>Rewards<br>Points<br>Incentives<br>Leader boards | |
| Feedback elements | | Calorie tracker<br>Step-counters<br>Distance travelled<br>Daily notifications<br>Time spent<br>Heart rate | Breathing rate<br>Sleep cycle<br>Sound inputs<br>Weight loss indicator<br>Physical form-checker<br>Gait/posture checker |

found that cooperative exergaming promotes self-efficacy, intrinsic motivation, pro-social behaviors, and continued gameplay, whereas competitive exergaming may promote short-term physiological arousal, and acute bouts of aggression. Furthermore, *Song et al. (2013)*, investigated the role of competitive and cooperative play on intrinsic motivation in the context of exergames and found that players who were highly competitive enjoyed playing competitive games, whereas players who were not competitive exercised just as hard as competitive individuals, but did not enjoy their experience. The researchers concluded that further research is needed to study the effects of competition on motivation in exergames and recommended that game design could tailor game-play to individual differences for increased exercise motivation.

Some researchers have adopted the idea of *personality-based tailoring* (*Mattheiss et al., 2017*; *Tondello et al., 2017*) and have conducted studies to explore the effects of adapting games based on competitive/cooperative preferences in game-play of an individual. For example, *Shaw et al. (2016a)* designed a virtual trainer system which adapts to a player's level of competitiveness/cooperativeness and found that the game was particularly enjoyable and motivating when the system aligned with the personality of the player. Additionally, *Chan, Whitehead & Parush (2017)* examined the effects of competitive and cooperative attitudes in paired play situations and found that competitive individuals prefer playing alone, whereas cooperative individuals enjoy playing together. To increase the level of enjoyment between player pairs, the researchers recommended that if player pairs favor the use of cooperative strategies for success, then offer cooperative game scenarios. In contrast, if player pairs favor the use of competitive strategies for success, then offer individual or competitive scenarios.

In a different study, *Hagen et al. (2016)* designed "Pedal Tanks," which is an online competitive multiplayer stationary bicycle exergame based on common team-based shooter games, as well as features from multiplayer online battle arena games.

The researchers evaluated the game and reported that the level of enjoyment, physical exertion, and interest were all higher compared to moderate pace walking. However, the study was only conducted with eight participants in three game sessions which is not enough to draw conclusions that the game is able to motivate long-term play. In a more recent study, the same research group reported that Pedal Tanks is highly enjoyable and is capable of encouraging high-intensity interval training (*Moholdt et al., 2017*). They concluded that this form of training can be a viable option for motivating those who have no interest in exercise. Consistent with the findings of *Hagen et al. (2016)* and *Moholdt et al. (2017)*, researchers such as *Monedero, Lyons & O'Gorman (2015)* also observed that competitive cycling videogames elicited higher levels of energy expenditure and were rated more enjoyable than conventional stationary cycling exercises.

In a related research, *Staiano, Abraham & Calvert (2013)* conducted a 7-months study to examine the effects of competitive and cooperative play on self-efficacy. Results of the study revealed that cooperative exergame players lost significantly more weight than the control group who gained weight over time and competitive exergame players did not lose weight. The researchers theorized that cooperation may foster a team bond experience more so than competition which could have motivated persistent play. Results also showed that adolescents who lost weight were more likely to have high initial levels of peer support which may promote group cohesion and provide social reinforcements that aid in sustaining play in the cooperative condition. The effects of competitive/cooperative behaviors in exergames remain a popular research topic for many videogame researchers and designers.

In summary, researchers have found that not all players enjoy competitive playing scenarios and rather than employing a "one-size-fits-all" approach (*Orji et al., 2013*) designers and researchers are attempting to personalize the exergaming experience, as well as conducting long-term studies to examine the effects of competitive and cooperative play.

## Situational interest

Situational interest refers to an interactive psychological state that occurs at the moment there is a match between a person and an activity (*Chen, Darst & Pangrazi, 1999*) and has been identified as a powerful motivator particularly in the educational context (*Hidi, 2001*). *Deci (1992)* has suggested that teachers, who promote student autonomy and choice in their classrooms, increase the level of intrinsic motivation and situational interest. Within the context of physical education, *Chen, Darst & Pangrazi (1999)* developed a multidimensional model of situational interest which consists of five sources: (1) novelty, (2) challenge, (3) attention demand, (4) exploration intention, and (5) instant enjoyment.

These sources have been studied in the context of exergames. For example, researchers such as *Pasco et al. (2017)* designed a mobile application-based exergame that can be paired wirelessly using Bluetooth technology to an exercise bike and evaluated its effectiveness for motivating moderate-to-vigorous levels of physical activity and generating greater levels of situational interest. Results showed that the mean scores for participants' physical activity levels, situational interest sources, and total interest differed significantly between the experimental group and the control group. The researchers concluded that their design is effective for promoting light levels of physical activity, situational interest,

and other important psychological determinants of physical activity participation compared to conventional biking exercise. In a different study, *Sun (2013)* examined the effectiveness of exergames for motivating physical activity during physical education class over a two semester period. A total of 70 grade 5 students rotated between eight different gaming stations to engage in a different full body movement games. Measurements were obtained to assess physical intensity and situational interest. Results revealed that the amount of physical activity increased overtime, yet interest declined. Additionally, boys reported higher levels of game enjoyment compared to girls, and both boys and girls reported lasting feelings of challenge, exploration, and instant enjoyment. It was concluded that although there is evidence that exergames may have strong motivational power at the initial stages of game play, yet whether or not it is able to sustain motivation in the long-term is questionable.

Overall, situational interest seems to be an effective strategy for designing exergames and increasing exercise intensity, but the long-term effects are unknown.

## Social interaction

From the HCI perspective, *Dourish (2002)* argues that physical and embodied features of interactive systems are related to the features of social settings. However, *Salen & Zimmerman (2013)* suggest that design alone cannot warrant social play—designers can offer features that facilitate social play, but it is the players who create it. *Bekker, Sturm & Eggen (2010)* explain that social interaction can be provoked by a game which is played by multiple players and the type of relationship between the players can influence the type of social interactions that occur. *Yee (2007)* developed an empirical model of player motivations in online games which identified "social interaction" as one overarching component which consists of three subcomponents:

1. Socializing: having an interest in helping and chatting with other players;

2. Relationships: the desire to form long-term meaningful relationships with others, and;

3. Teamwork: deriving satisfaction from being part of a group effort.

These sub-components provide a foundation for conducting quantitative research in online games to better understand usage patterns, in-game behaviors, and demographic variables such as age and gender, with respect to player motivations.

In a videogame context, research shows that social interaction is important for experiencing enjoyment[2] (*Sweetser et al., 2017*; *Sweetser & Wyeth, 2005*) and exertion games can facilitate social play in computer mediated environments (*Mueller, Gibbs & Vetere, 2009*, *2010*). Research also shows that social presence (*Ekman et al., 2012*), social connections (*Przybylski, Rigby & Ryan, 2010*), social benefits (*Granic, Lobel & Engels, 2014*), and the experience of social relatedness (*Kooiman & Sheehan, 2015b*) are common motivations for videogame play. Furthermore, research in videogames suggest that social presence is an important determinant of player enjoyment (*Gajadhar, De Kort & Ijsselsteijn, 2008*) and mediating factors drawn from behavioral health theory such as communication and social support can be impactful methods for designing videogames to encourage exercise behaviors (*Lieberman et al., 2011*). In the context of exergames, cooperative play increases social interactions (*Park et al., 2012*) and engaging in social

[2] Other components of flow include concentration, challenge, skill, control, goals, feedback, and immersion. *Sweetser & Wyeth (2005)* suggest that social interaction is not an element of flow, but is a strong element of enjoyment in games. The researchers further explain that social interaction is not a property of the task as are the other elements of flow, but the task is a means to allow social interaction.

**Table 4 Design guidelines for social exertion games (*Mueller, Gibbs & Vetere, 2009*).**

| | Theme | Design implications |
|---|---|---|
| 1 | Exertion and meaning | Networked exertion games should support meaning making through exertion by supporting various levels of intensity of the game activity. |
| 2 | Metagaming | Metagaming is a social play phenomenon that refers to the relationship of a game to elements outside of the game and can be supported by allowing distributed participants to retell the exertion activity with their body. |
| 3 | Synchronized exertion | To facilitate the synchronicity between players, offer opportunities for kinesthetic movements. |
| 4 | Uncertainty | Exertion can amplify the level of uncertainty afforded by tangible objects. |
| 5 | Breaking the rules | Players might reshape the game structure with their actions, with positive and negative consequences. |
| 6 | Awareness and exertion | A mismatch of awareness space and exertion space can influence social play. The exertion space might be larger than the game-play, as players move around during and in-between games. |

interactions, during exergame play is an important experience leading to increased immersion between players (*Lee et al., 2017*). There is also some research suggesting that pre-existing social relationships (e.g., friends) are not enough to sustain long-term motivation for physical activity and suggest that increasing game complexity and variations can help to encourage long-term exergame play (*Caro et al., 2018*). Recently, *Kaos et al. (2019)* conducted a 6-week study and found that social play have superior adherence compared to individual play because based on SDT (*Deci, 1992*), players who engaged in social play allowed them to experience a sense of *belongingness*—a universal human need to form and maintain relationship with others (*Baumeister & Leary, 1995*), was satisfied.

*Mueller, Gibbs & Vetere (2009)* studied the potential of exertion games for facilitating social play in networked environments. The researchers conducted a qualitative analysis of one particular case study on 39 participants in 13 distinct teams of three players and identified six salient themes on how exertion games can facilitate social play (Table 4). Games designers who are interested in designing games which offer social interaction can use these six themes as a guide on how to include exertion aspects, as well as exertion applications that currently do not support social play.

In summary, designers and researchers are interested in developing playful experiences that support social interactions. Social support and communication are key mediators for encouraging behavior change and continued play. However, pre-existing social relationships between players are insufficient for sustained long-term play. Games that offer a variety of challenges and gradual release of game features can encourage longer-term play.

## GAP ANALYSIS

### Summary of findings

To summarize, using the snowball method, we conducted a review on some of the most common theories and elements that researchers have studied in the exergaming literature. Our aim was to search for motivators that particularly enhance long-term play. We identified that the SDT, gamification, competition and cooperation, social interaction, and situational interest are the main categories that represent the most active approaches in existing research in this area (Table 5). We also found that the social aspect

**Table 5 An overview of selected studies examining the effectiveness of motivational strategies and approaches particularly for exergame adherence over the long-term.**

| Reference | Exergaming activity | Activity duration | Outcome measure | Results | Limitations |
|---|---|---|---|---|---|
| **Self-determination theory** | | | | | |
| *Limperos & Schmierbach (2016)* | Biggest loser—skate or splash (Nintendo Wii) | 30 s to 10 min | Enjoyment, competence, autonomy, presence, future intention | Better performance is associated with great levels of autonomy, competence, presence, enjoyment and future intentions to play. | Sample population: College students only; not longitudinal study. |
| *Lin et al. (2012)* | Walking, running, jumping | 15 min | Enjoyment, effort for gameplay, motivation for future play, self-efficacy | Autonomy-supportive and competence-supportive game features leads to greater game enjoyment and motivation for future play, whereas satisfaction supportive game features leads to more energy expenditure. | Studied only group of game features, not individual game features. Gamers only, do not apply to non-gamers. |
| *Osorio, Moffat & Sykes (2012)* | Wii Sports, DDR | N/A | Motivation with respect to SDT | Enjoyment of social interactions is a major factor for playing exergames. | Small sample size, university students only. |
| *Staiano et al. (2016)* | Group-based dance exergames—for example: Just Dance | 60 min sessions, 3 times a week for 12-weeks | Physical Activity, self-efficacy and intrinsic motivation | Intervention group reported higher levels of intrinsic motivation and improved self-efficacy toward physical activity compared to the control group. | Limited to obese female adolescents. |
| **Gamification** | | | | | |
| *Zhao et al. (2017)* | Walking, running, cycling | 70 days—ranging from 4 to 6 min/session | Time, satis-faction, encourage-ment, enjoyment | Gradual release of game features is effective at keeping players engaged. | Sample population: university students only; environment: indoors only. |
| *Geelan et al. (2016)* | Augmented cycling and rowing exercises | 30–90 min | Energy expenditure, physical intensity, duration and enjoyment | Augmenting traditional exercise equipment with gamified elements can increase the time spent exercising, when compared to non-gamified exercise equipment. | Small sample size, short-term. |
| **Competition and cooperation** | | | | | |
| *Staiano, Abraham & Calvert (2013)* | Nintendo Wii sports games | 20-weeks, 30-60 min per day | Changes in weight, peer support, self-efficacy, and self-esteem | Cooperative game is more effective than competitive game for weight loss. | Studied only children (15–19 years old); small sample size; only one measure of body adiposity (weight). |
| *Chan, Whitehead & Parush (2017)* | Virtual bocce | 1 h | Enjoyment and social influence | Competitive pairings enjoyed competitive game scenarios, whereas cooperative pairings enjoyed cooperative game scenarios. | Small sample size; unbalanced gender and attitude pairings. |
| *Song et al. (2013)* | Hula Hoop, a Nintendo Wii Fit program | 10–18 min | Intrinsic motivation, mood, exercise efficacy, competitive-ness, expected competence, heart rate | Highly-competitive individuals enjoy competitive game settings, whereas non-competitive individuals do not enjoy competitive game settings. | Short-term study; manipulation of competition context was limited to employing external factors (e.g., a reward system) rather than changing the exergame content itself. |

| Reference | Exergaming activity | Activity duration | Outcome measure | Results | Limitations |
|---|---|---|---|---|---|
| *Shaw et al. (2016b)* | Virtual cycling | Three 10 min exercise sessions | Enjoyment and motivation | Competitive trainer profile is motivating for competitively inclined individuals and cooperative trainer is more motivating for goal oriented individuals. | Small sample size; short pilot study. |
| *Peng & Crouse (2013)* | Space-Pop Mini game (Kinect Adventure) | Two rounds —2 min and 20 s. | Enjoyment and motivation | Parallel competition in separate physical spaces was the optimal mode, leads to both high enjoyment and future play motivation and high physical intensity. | Cooperation in different physical space and competition in the same physical space was not studied. |
| Situational interest | | | | | |
| *Pasco et al. (2017)* | Stationary biking | 15 min | Situational interest and physical activity levels | A mobile application-based exergame capable of being wirelessly paired to an exercise bike can promote light physical activity, situational interest, and other important psychological determinants of physical activity participation compared to traditional biking exercise. | College students only; short-term study; game not challenging enough for highly fit individuals. |
| *Sun (2013)* | Wii Sports—dance games | Twice a week and each lesson was 30 min long | Enjoyment | Prolonged exposure to exergaming activities may lead to decreased perception of situational interest, which might lead to lower motivation to engage in exergames-based physical activity in the future. Boys perceived exergaming experience to be more enjoyable than girls. The use of a variety of types of games might result in enhanced physical activity intensity over time. | Results based only on children; students were required to participate in the exergaming activities as part of the physical education program rather than the opportunity for choice, and thus, limited autonomy. |
| Social interaction | | | | | |
| *Mueller, Gibbs & Vetere (2010)* | Table tennis | 30–60 min | Playing experience | Anticipation and accountability are key themes for the design of social play specific to exertion games. | Limited to one particular game; based only on qualitative observations and analysis. |
| *Park et al. (2012)* | Running on a treadmill | Four 20-min sessions over 2 weeks | Enjoyment, energy expenditure, competition and cooperation | Cooperative play increases social interaction—there is a relationship between individual willingness, derived intrinsically or from the relation between co-players, and cooperation and the cooperation experience. | Small sample size; 20–25 year olds only; no features to prevent overexertion. |
| *Kooiman & Sheehan (2015b)* | Virtual bowling and virtual table tennis | 40 min | Intrinsic motivation —relatedness | Motivation is high in online and non-player character exergaming contexts. | Results based on children only (ages 11–18) and short-term. |

(Continued)

| Reference | Exergaming activity | Activity duration | Outcome measure | Results | Limitations |
|---|---|---|---|---|---|
| *Kaos et al. (2019)* | Networked, cycling-based exergame | 6 weeks | Win rate, time as indicator of engagement, total physical activity and belonging-ness. | Players who actively engaged in social play had significantly higher exergame adherence compared to players who primarily played alone. | An exploratory study based on post hoc analysis of data collected in an earlier study and lack of personality assessment. |

of an activity and social experiences are important motivators for continued exergame play. An interesting trend that we noticed was that many game designers and researchers are gradually moving away from designing for and researching exercise games that are confined to indoors and require gaming consoles. Increasingly, they are exploring the effectiveness of exergames that are app-based for outdoor play and researching exergame experiences where exercises occur outside of play sessions. Another trend that we gathered was rather than designing based on a "one-size fits-all" approach, many designers and researchers are beginning to explore the effects of a more tailored approach to better satisfy individual and group differences, as well as personalize the game experience.

## Discussion

Current literature suggests that exergames can be an effective tool for increasing exercise motivation and are generally rated more enjoyable compared to traditional forms of exercises. While some research suggest that commercially available off-the-shelf videogames are capable of providing light-to-moderate levels of physical activity (*Peng, Crouse & Lin, 2013*), other research shows that due to inconsistent results and the overall poor methodological quality of some studies, whether or not exergames are suitable to meet physical activity guidelines for different population groups are inconclusive (*Höchsmann, Schüpbach & Schmidt-Trucksäss, 2016*). There is also some research suggesting that the physical demand of some exergames are not intense enough to contribute toward the recommended daily amount of exercise for children because these games do not require the same level of energy expenditure as the actual sport (*Graves et al., 2007*). Exergames can be designed to adjust intensity. For example, to encourage adequate energy expenditure, *Whitehead et al. (2010)* propose that exergames must consist of two core features: (1) rewards for encouraging long-term motivation and, (2) better features or game-play that will involve full body movements. To prevent overexertion, *Schneider & Graham (2017)* investigated the use of "nudges" (*Thaler & Sustein, 2017*) and found that nudges motivated players to slow down, and fit naturally into the games.

To promote longer-term play, designers and researchers are investigating various strategies and mechanics in addition to the game itself such as gamifying the exercise experience (*Matallaoui et al., 2017*; *Mueller et al., 2011*) and personalizing the experience (*Göbel, Hardy & Wendel, 2010*), cultivating group cohesion using smart watch interactions

(*Esakia et al., 2017*), augmenting traditional exercise equipment with gamified elements (*Geelan et al., 2016*), adapting content presentation (*Dias & Martinho, 2011*), increasing situational interest (*Pasco et al., 2017*; *Sun, 2013*), tailoring virtual coaching to individual personalities (*Shaw et al., 2016a*) and many others. Research shows that experience of fun and social affiliation are good predictors of long-term intention to playing exergames (*Adam & Senner, 2016*). Player achievement predicts feelings of autonomy, competence, presence, enjoyment, and continued motivation to play exergames (*Limperos & Schmierbach, 2016*). Although informative, there could be other elements that could encourage longer-term play that remain unexplored. More research is needed to understand what individual (e.g., personality dimensions and motivational orientations), social (e.g., competition vs. cooperation) and cultural (e.g., individualistic vs. collectivistic preferences) factors which might or might not encourage continued exergame play.

In addition to the elements and theories that we presented in this literature review, we recognize that there are many other factors, theories and elements that can make a game attractive and fun to play. For example, other elements that make games compelling include opportunities to master a task or learn and practice a new skill (*Koster, 2013*), construct, explore, and interact with the game-world environment (*Moore & Sward, 2006*), as well as experience a fictional world (*Juul, 2005*). However, we believe that the elements and theories we reviewed in this survey paper are particularly useful for understanding retention of exergames.

## Active research questions and directions for future research

Many questions remain unanswered regarding how we can explore and apply these motivational theories and elements that encourage long-term play. Below are some unexplored research questions provoked by the discussion above as researchers continue to investigate theories and elements for encouraging exergame adherence over the long-term. We offer three general areas for further investigation: (1) enrichment of the social experience and tailoring to the individual, as well as, group personality, (2) development of metrics specialized for evaluating the exergaming experience, and (3) conduct longitudinal studies to better understand the effects of competition, cooperation, and situational interest on exergame adherence over a longer period of time.

### Social experiences

More research is needed to uncover optimal player tailoring and pairing mechanisms for facilitating group play. Research shows that in a competitive exergame context, the level of intrinsic motivation increases for players who are highly competitive yet decreases for players who are less competitive (*Song et al., 2013*). It is important to tailor the game experience based on individual preferences to attract all player types. *Chan, Whitehead & Parush (2017)* found that matching game scenario and individual attitudes to a player's competiveness and cooperativeness increases the level of enjoyment. However, the effects of attitude pairings in larger groups and the long-term effects of competitive and cooperative play are still unknown. Social exergames are popular because they offer

competitive and cooperative experiences that are similar to group exercise (*Marker & Staiano, 2015*) and supportive peer relationships encourage exercise adherence (*Murcia et al., 2008*). However, current exergame research seems to focus mainly on designing for single player (*Geelan et al., 2016*; *Shaw et al., 2016b*) rather than multiple players. Although some researchers (*Choi et al., 2016*; *Esakia et al., 2017*; *Zhao et al., 2017*) are investigating the potential of wearable devices for encouraging participation in physical activity and positive group cohesion, there is not enough research examining the effects of social interaction and social connectedness in larger groups. Additionally, although some researchers are beginning to study the effects of group play and have found that group exergaming can provide enjoyable physical activity experiences (*Staiano et al., 2018*), there is very little research on how to form groups that motivates continued play.

 Potential Research Questions:

1. What social elements enhance group play?
2. What exergame features can be offered to enhance feelings of social affiliation for increasing adherence to long-term play?
3. How can an exergame be designed to offer a more cohesive group experience?
4. How can the issue of small sample size in the majority of studies be addressed?
5. What design improvements can be made to personalize the exergame experience to increase the effectiveness for multiple players?

### Personality-based solutions

In social settings, people can sometimes get along well and build positive relationships but other times they can remain distant. Research across a number of contexts of human social activity has shown that the compatibility between people depends on the degree to which their personalities are similar (*Tenney, Turkheimer & Oltmanns, 2009*). It is likely that situations, where people get along well, could be because they share common interests and have similar personalities. Alternatively, situations, where people remain distant, could be due to lack of common interests and personality clashes. Personality (a set of individual attributes that consistently differentiate persons from each other in the ways they think, feel, and act (*Ones, Viswesvaran & Dilchert, 2005*)) has been shown to play an important role in many areas of HCI including user-interface design (*Nov & Arazy, 2013*), gamification (*Codish & Ravid, 2014*; *Orji, Tondello & Nacke, 2018*), and video games (*Braun et al., 2016*; *DeGraft-Johnson et al., 2013*). In exergames, *Mattheiss et al. (2017)* examined the personality of Pokémon Go players on motivation to continue play and found that players who play the game score low on conscientiousness, whereas players who stopped playing the game after 3 months score higher on neuroticism. This is an important finding because as researchers continue to explore the relationships between social exergame play and personality, understanding which personality types are more or less likely to continue play can help designers tailor the social experiences to satisfy individual players, but also groups of players.

The literature about personality-based tailoring and matching in exergames is still in its infancy. The results of some studies (*Orji, Nacke & Di Marco, 2017*; *Shaw et al., 2016a*) suggest that a tailored approach, particularly in the context of persuasive games and gamified systems, are more effective at motivating health-related behaviors compared to a non-tailored approach. However, the long-term effects of these approaches still remain unknown.

Potential Research Questions:

1. Do different personalities prefer different types of rewards?
2. Does personality composition of groups play a role in exergame retention?
3. What personality compositions are more likely to encourage continued play and which ones do not?

### Metrics for evaluating the exergaming experience

In a narrative review, *Mellecker, Lyons & Baranowski (2013)* proposed that validated personality tests should be applied to exergame play for experiencing enjoyment. To date, most commonly used metrics for evaluating player experiences such as the level of enjoyment is a subscale of the Intrinsic Motivation Inventory (*McAuley, Duncan & Tammen, 1989*), which is a questionnaire based on the SDT or the 10-item Positive Affect/Negative Affect Scale (*Watson, Clark & Tellegen, 1988*), social presence—the Social Presence Gaming Questionnaire (*De Kort, IJsselsteijn & Poels, 2007*), engagement—the User Engagement Scale (*Wiebe et al., 2014*). Yet, none of these instruments were specifically developed for evaluating player experiences in an exergame context. New metrics not only need to be developed for evaluating personality, but also other factors which could specifically influence the player's exergame experience. *Lee et al. (2017)* identified that one major limitation in exergaming research is that most studies to date rely on measures developed for either normal exercise or sedentary digital games. Thus, the researchers recommend that a unique evaluation for measuring the psychological effects of exergames is needed because exergames are different from sedentary video games and traditional kinds of exercise. Exergames consist of features such as the level of exertion, which might not be present in other games such as sedentary videogames and are important aspects of creating an enjoyable experience (*Lee et al., 2017*). Another limitation with respect to assessing player experiences in exergames is that majority of research relies on survey reports and subjective data. More research should aim to gather objective measures such as heart rate or galvanic skin response because they provide good validation for subjective reports in a co-located, collaborative play environment (*Mandryk & Inkpen, 2005*).

Potential Research Questions:

1. How can scales be improved to better measure outcome variables that target the motivational effects of exergames?
2. What other measures can be collected to give a better picture of exergame interest, enjoyment, and engagement?

### Call for longitudinal studies

Finally, longer-term studies are also needed in the realm of exergames. *Macvean & Robertson (2013)* conducted a 7-week study on user's physical activity, motivation, and behavioral patterns on using exergames, and suggested that longitudinal studies are necessary for evaluating motivational effects, since it ensures that the intensity of a user's behavior is appropriate and sustained. Likewise, *Marker & Staiano (2015)* highlighted that no studies have tracked changes in physical activity with respect to social exergaming in competitive vs. cooperative contexts long-term. Long-term studies are important, particularly for exergames, because if aspects of the game (e.g., technology, other players, content presentation) is inadequate for sustained play, game designers can refine game elements to ensure optimal playing experience.

Potential Research Questions:

1. What are the long-term effects of competitive and cooperative in social exergaming?
2. Is situational interest effective at motivating continued exercise interest in exergames over the long-term?

## CONCLUSIONS

In closing, many researchers and game designers are exploring various technological solutions to reduce sedentary behaviors and leverage interest for sustained physical activity. One possible solution is exergames because they can make exercise seem more enjoyable and engage players. Research shows that exergames are effective at motivating people to participate in exercise. However, simply adding game elements to an activity or gamifying exercise routines may not necessarily increase the level of motivation and achieve the desired outcomes. Research also suggested that exergames may not be able to motivate exercise interest over a long-term and some games are inadequate for encouraging intense physical activity that are necessary to reap the full health benefits.

In this review, we identified that SDT is one of the most commonly used theories to inform the design and evaluation in the exergame literature. In particular, satisfying the need to belong is an important component for designing multiplayer exergames because it increases the likelihood of game adherence. Principles of gamification and motivational affordances have been applied in attempt to make participation in traditional exercise more interesting. The results of these studies show that gamifying the exercise experience can lead to increased enjoyment, exercise time, and physical exertion. Furthermore, we observed that recent technologies such as virtual reality combined with gamification features can easily be added to traditional exercise equipment such as a stationary bike can make the exercise experience more interesting. Competition and cooperation are also popular for motivating exergame play. We came across many studies showing that some players are motivated by competition while others are motivated by cooperation and when designing exergames, tailoring scenarios based on one's competitive/cooperative preferences can increase the level of game enjoyment. Offering opportunities for social interaction between players can also enhance exergame play, and including elements of

situational interest can increase the level of exertion, as well as make the exercise activity more enjoyable. However, there were studies that evaluated the long-term effectiveness of situational interest on exergame adherence and found that prolonged exposure to exergaming activities could lead to decreased perception of situational interest and in turn, could lower motivation to participate in exergames-based physical activities in the future.

Finally, as online exergames become more popular, there will be a wide range of possible player-pairings. This creates an opportunity for people to be brought together dynamically to play exergames and engage in social interactions in a way that might increase or decrease the level of motivation among multiple players. Thus, it is important to understand how these pairings can motivate interest and encourage long-term participation beyond the novelty of a new game or technology keeping players highly engaged and motivated so that they keep on returning to the game, and ultimately continue an exercise plan, as well as an active lifestyle.

## ACKNOWLEDGEMENTS

This paper is dedicated to the memory of our friend and colleague, Dr. Anthony Whitehead, who contributed to the planning and defining the early stages of this work.

### Funding

The authors received no funding for this work.

### Competing Interests

The authors declare that they have no competing interests.

### Author Contributions

- Gerry Chan conceived and designed the experiments, analyzed the data, prepared figures and/or tables, authored or reviewed drafts of the paper, approved the final draft.
- Ali Arya conceived and designed the experiments, authored or reviewed drafts of the paper, approved the final draft.
- Rita Orji conceived and designed the experiments, authored or reviewed drafts of the paper, approved the final draft.
- Zhao Zhao conceived and designed the experiments, authored or reviewed drafts of the paper, approved the final draft.

### Data Availability

This is a review paper, not an empirical study. A list of all 30 articles that are included in the analysis are available in the Supplemental Files. It was used to generate means and standard deviation scores for citation counts.

This is a review paper, not an empirical study.

## Supplemental Information

Supplemental information for this article can be found online at http://dx.doi.org/10.7717/peerj-cs.230#supplemental-information.

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
