# Peer review of "Motivational strategies and approaches for single and multi-player exergames: a social perspective"

_PeerJ Computer Science, doi:10.7717/peerj-cs.230_

## Round 0.1 · original submission · Minor Revisions

The reviews list a number of aspects that need to be improved before publication. A partial list of such aspects includes:

1) a clarification of the overall number of papers reviewed for this survey. For example, Reviewer 2 interprets the 30 papers supposedly selected as seed for the snowball technique as the entire set of the papers reviewed, which indeed would be too narrow for this topic.

2) clarification of some terms (such as "retention" and "sustainability", as suggested by Reviewer 2) and a crisper articulation of ideas (as Reviewer 1 suggests throughout the entire review).

3) a comparison with other related literature reviews, as to crearly address the merit criteria of the journal. Specifically, please answer in your narrative the following: "Has the field been reviewed recently? If so, is there a good reason for this review (different point of view, accessible to a different audience, etc.)?" (from Basic Reporting literature review PeerJ criteria).

Reviewer 1 ·

Basic reporting

This paper is a generally well-written literature review of motivational strategies in exergames. The language used throughout is of a high standard with few grammatical errors or typos. Claims made by the paper are supported by references to the literature.
The paper is logically structured, though I question the placement of tables and figures after the references as opposed to at the relevant locations in the body of the paper. Perhaps this is to do with the review manuscript format? The tables and figures are clear and well laid out, though figure 2 does not appear to provide any information that is not already present in the text. Given that figure 2 is taken from cited work, I question whether the paper is improved by the presence of this figure. Relevant raw data is shared.

Table 1 has entries for what appears to be three sets of table notes, but only one note is provided, that a duplicate of text provided above the table.

This review is relevant to the scope of the journal, and of interest both within the disciplines of computer science and health. Although there have been other reviews related to exergames in the last few years, the particular focus of this paper on motivational strategies makes this paper sufficiently distinct from these existing reviews.

The paper is well motivated, with the retention problem of exergames clearly identified, justifying the review topic. However, I note that the background section of the abstract makes an unclear claim related to this introduction. Line 18 states “[exergames] have been shown to address a variety of health-related issues.” As it stands, this reads like exergames offer health benefits beyond those given by the increase in exercise levels. If this is what the authors are claiming, this statement needs to be supported by literature. If the authors are just claiming that the benefit lies in increasing exercise levels, this sentence should be reworded to make that clear.

Experimental design

This paper falls within the aims and scope of the journal.

This paper follows a standard approach for conducting a literature review, with a database search and snowballing from an initial set of papers. However, reporting of the methodology would benefit from some clarification.

The exact terms the authors use in their database search are unclear. It reads as if the search terms listed on lines 111 and 112 were searched independently, but given the numbers reported, this seems unlikely. My own search for the term ‘active video games’ gives many more results than the authors report, or many fewer if searched in quotation marks.

The databases selected for searching should provide an appropriate coverage of the relevant literature. Was article full text searched for the terms, or just titles and abstract?
The authors also mention that the search included reference sections of previously gathered articles, but do not mention how those articles were chosen.

Line 133 states that the majority of journal articles have a high citation count. Is this relevant, and if so, is a mean with such a high SD the best way to report this fact?

Sources are thoroughly cited with relevant quotations. However, there are cases where papers are discussed but the relevance of the points identified is unclear. For example in lines 592 to 595, the authors identify specific traits associated with two different levels of adherence to playing a specific game, but the authors do not effectively connect this to the rest of the discussion in the paragraph.

The review is sensibly organised into appropriate sections and subsections.

Validity of the findings

The authors’ conclusions are clearly stated and supported by their findings in their survey of the literature. However, the conclusion section could better summarize the items discussed through the paper: the author’s discussion is broad but the summarization is brief.

Gaps in the literature and research questions for future work in the field are clearly identified. Section 3.3.1 groups social experiences and personality-based solutions as a single area, but although some personality-based solutions in the literature happen to target social experiences, these topics are not inherently related. As such, this section might be improved by separating these concepts.

It is strange to identify gamification as a motivational element in exergames. Rather, gamification is an inherent property of exergames: an exergame without gamification is no longer an exergame, merely exercise.

Additional comments

Overall this is a good paper, though a few things need to be reworded or clarified before publication.

Below is a list of formatting minor errors and odd phrasings that I identified:

There appears to be some inconsistencies in citation style within the text. In some cases, multi-author papers are listed as First Author et al., while in other cases the full set of authors are listed. An example is the contrast between the citations on lines 472 and 473, but this happens throughout the text. Additionally, on line 572, a paper is cited by the first author’s full name, rather than their surname.

There is an unnecessary comma on lines 403, 509, 685.

Line 242 “the grow strength of the avatar” this phrase doesn’t make sense.

Line 282 “abundant” should be “abundance”

Line 471 “In the context exergames” -> “In the context of exergames”

Line 555 “tailor to individual” doesn’t make sense. Maybe “tailoring to the individual”?

On line 632, competitive and cooperative social elements are listed as items that “might not be present in other games such as sedentary video games or online games”. There is no reason identified to believe that exergames are any more likely to contain these elements than sedentary video games, and I would expect that online games are in fact more likely to contain these elements than any other sort of game.

Lines 601 - 603 make a claim that certain topics are “on the research agenda for many researchers”. Although this is certainly likely to be true, it’s difficult to prove without a survey of the researchers involved and the section would be stronger without this claim.

Reviewer 2 ·

Basic reporting

The general reporting of the literature review is of satisfactory quality. The authors demonstrated a good command of language. The paper is well-written and easily understood. The topic of sustainability of exergame is of interest to the PeerJ CS community.

Experimental design

The study design is unsatisfactory. Plenty of papers have been published in this area and the authors only identified 30 papers based on their criteria which is shocking. The paper should rather be more focused on a subset of the area such as sociality that was later brought up in the discussions. The paper reads like a literature review done for a MS thesis and is entirely inadequate for publication.

Validity of the findings

One of the biggest drawback is that the paper failed to define what the authors mean by "sustainability" as the paper primarily focuses on the "retention problem". Does retention mean sticking to the same exergame for a prolonged period of time? What if the players moved onto another game? How long is considered to be "sustainable"? The authors kept on referring to the intended outcomes of the surveyed research papers using the words "physical activities" and "exergames" interchangeably, but it isn't clear why those two are even at all correlated with one another. Does playing more exergames mean increased physical activities in the general sense? It seems that the authors fundamentally confounded these basic concepts from the onset, and the research is build on flawed premises. Also, the sections also seem rather arbitrary. Are competitive and cooperative elements of an exergame features of "gamification"? How would the authors differentiate situational interests from other contextual factors that affect a player's motivation? The literature is scattered and fragmented and no new insights is generated from the way the authors synthesized the 30 papers. The gaps section also doesn't build on the literature review at all. Personality traits is something that has been investigated in previous gaming literature which were curiously left out of the 30 papers reviewed by the authors.

Additional comments

It'd be wise if the authors would focus their review on a smaller, more identifiable, and more focused research problem of exergame, and perhaps conduct a small empirical study to demonstrate the intended claims. As of now, the scope seems to be beyond the authors' abilities to handle and this work is not of publishable quality.

---

## Round 0.2 · accepted · Accept

The revised version has addressed the main concerns of the reviewers. There are minor edits suggested in the latest review that the authors will want to consider for the published version while in production.

Reviewer 1 ·

Basic reporting

See general comments.

Experimental design

See general comments.

Validity of the findings

See general comments.

Additional comments

I am largely satisfied with the changes the authors have made to the paper since my original review. Below are some further grammar errors and issues with choice of language that I suggest are corrected before publication, without need for re-review.

Line 57:

"maintaining players motivation and keeping actively engaged" change to "maintaining players' motivation and keeping them actively engaged"

Lines 60-62:
Use of terms retention and sustainability interchangeably is questionable, unless there is a good reason for it such as frequent use of both terms in the literature, I would prefer the authors just use retention as sustainability holds some different implications.

Line 67: "the participation" change to "the word participation"

Line 77 "exist" -> "exists"

Line 126: If the listed databases are all of the databases used, the word "including" is unnecessary. If they are not, then all databases used in the search should be used.

Line 180 "particularly focused" -> "we focused" or "we particularly focused"

Line 692: Differentiation of sedentary videogames and online games is an odd choice for contrasting with an exergame. Both sedentary and non-sedentary games can be online.

Line 733: "We learned" is not an appropriate choice of words.

Line 744 "decrease" -> "decreased"